# Bone-Differentiation-Associated Circ-Spen Regulates Death of Mouse Bone Marrow Mesenchymal Stem Cells by Inhibiting Apoptosis and Promoting Autophagy

**DOI:** 10.3390/ijms25053034

**Published:** 2024-03-06

**Authors:** Ziwen Liang, Bingjie Luo, Bojia Peng, Yunchuan Li, Xueling Hu, Wenqiang Zhong, Xiaoyun Li, Panpan Wang, Xiaofeng Zhu, Ronghua Zhang, Li Yang

**Affiliations:** 1College of Pharmacy, Jinan University, Guangzhou 510632, China; 2Guangdong Provincial Key Laboratory of Traditional Chinese Medicine Informatization, Guangzhou 510632, China; 3College of Traditional Chinese Medicine, Jinan University, Guangzhou 510632, China

**Keywords:** *ERβ*, Circ-Spen, apoptosis, autophagy, mBMSCs

## Abstract

The role of estrogen receptor β (*ERβ*) in bone health is closely associated with its function in vivo, and *ERβ*^−/−^ mice have been widely utilized to explore the related influences. In this study, *ERβ*^−/−^ female mice were established to investigate the differential expression of circular RNAs (circRNAs) by RNA-Sequencing (RNA-Seq). Among these circRNAs, mmu_circ_0011379 (named Circ-Spen) exhibited high expression in *ERβ*^−/−^ female mice. However, the precise mechanism by which Circ-Spen regulates bone health remained unclear. This study identified Circ-Spen as a positive regulator of mouse bone marrow mesenchymal stem cell (mBMSC) viability. The expression of Circ-Spen was markedly increased in *ERβ*^−/−^ mice femurs tested by RT-qPCR. Moreover, Circ-Spen exhibited an enhanced expression during the bone formation process of mBMSCs. Qualitative experiments also demonstrated that Circ-Spen possessed a circular structure and was localized within the nucleus of mBMSCs. Functionally, it inhibited apoptosis via caspase-3, BCL-2, and BAX, while also promoting autophagy through BECN1 and P62 in mBMSCs tested by MTT assays, transmission electron microscopy (TEM), and Western blotting. These findings reveal the potential of targeting Circ-Spen as a promising therapeutic strategy for rejuvenating senescent mBMSCs and enhancing the efficiency of mBMSC transplantation, which lays the foundation for advancements in the field of bone therapy.

## 1. Introduction

Estrogen receptor β (*ERβ*) is one of the crucial receptors in women that mediates estrogen in the development and maintenance of bone, which is predominantly expressed in trabecular bone [1]. *ERβ* knockdown C57BL/6 mice have been widely used to research the effects of *ERβ* on bone and its mechanisms [2]. It has been reported that deleting *ERβ* protected both the cancellous and endocortical compartments from age-related bone loss by reducing the resorption and turnover of bone in aged female mice [3]. Furthermore, *ERβ* was found to be a key part of estrogen regulation of pluripotency and senescence in rat bone marrow mesenchymal stem cells (rBMSCs) [4]. During bone formation, BMSCs gradually differentiate into Osteoblasts (OBs), which is important in the synthesis, secretion, and mineralization of bone matrix [5]. In recent years, BMSCs have become the most commonly utilized seed cells in bone tissue engineering (BTE) due to their multifunctionality, which includes the secretion of a range of proteins with beneficial effects on surrounding cells and the promotion of bone tissue regeneration through their osteogenic, angiogenic, and immunoregulatory effects [6]. However, the survival and efficacy of BMSCs are significantly limited due to the substantial apoptosis and senescence that occur in the microenvironment following transplantation [7]. In essence, the regulation of BMSC activity is a crucial factor in enhancing transplant efficacy.

Circular RNAs (circRNAs), a type of non-coding RNA, possess a covalent bond linking the 3′ and 5′ ends through back-splicing. This unique characteristic enables them to regulate gene expression by blocking microRNAs, regulating transcription, and interfering with splicing [8]. Notably, substantial evidence has been proved for the differential expression of numerous circRNAs during the differentiation of BMSCs, with some of them playing a regulatory role in this process, underscoring their potential as novel biomarkers of stem cell differentiation and targets for stem cell-based therapies [9]. For instance, it has been reported that the expression of circular FKBP5 is upregulated during the osteogenesis of human BMSCs (hBMSCs), and dexamethasone modulates its expression during differentiation [10]. Additionally, circ-DAB1 enhances cell proliferation and osteogenic differentiation of hBMSCs by sequestering microRNAs (miRNAs) miR-1270 and miR-944, leading to the increased binding between RBPJ and DAB1 [11]. Similarly, circHGF is significantly upregulated in hBMSCs from patients with steroid-induced osteonecrosis of the femoral head, which targets the miR-25-3p/SMAD7 axis, leading to a suppression of both proliferation and osteogenic differentiation of hBMSCs in vitro [12].

The spen family transcriptional repressor (*Spen*) has been identified as a molecular integrator that facilitates the initiation of X-chromosome inactivation by linking *Xist* RNA with the transcription machinery as well as nucleosome remodelers and histone deacetylases at active enhancers and promoters in vivo [13]. It is not only necessary for the establishment of the Xi, but also crucial in the initiation of the XCI process [14]. *Spen* has previously been associated with different roles in different cancers [15,16]. Moreover, *Spen* plays a critical role in the cardiac function of zebrafish, mediated by molecular interactions with Connexin 43 [17]. However, the study of Circ-Spen was not conducted.

In this research, *ERβ*^−/−^ female mice were established to investigate the differential expression of circRNAs. Among them, the expression of Circ-Spen during osteogenesis in mBMSCs has been initially examined. Furthermore, the effect of Circ-Spen on the apoptosis and autophagy of mBMSCs as well as its underlying molecular mechanism were also investigated.

## 2. Results

### 2.1. The Differential Expressions of circRNAs in ERβ^−/−^ Mice

Since *ERβ* is closely related to bone health, wild-type (WT) and *ERβ* knockdown mice were established, and RNA-Sequencing (RNA-Seq) was used to discover the differential expression of circRNAs. Significantly, the expressions of five circRNAs were different in the two groups (Figure 1A). Among them, Circ-Hnrnpll and Circ-Spen were up-regulated in *ERβ*^−/−^ mice while Circ-Eed, Circ-Strn3, and Circ-Rabep1 were down-regulated. Their expression in WT and *ERβ*^−/−^ mice femurs was further validated by qRT-PCR, four of which were consistent with the result of the RNA-Seq, except for Circ-Eed (Figure 1B). Due to the great correlation and innovativeness of the Circ-Spen, subsequent experiments would be centered on it.

### 2.2. Identification of mBMSCs

Previous studies have suggested that CD29 and CD44 could be used as positive biomarkers, while CD11b and CD45 could be used as negative biomarkers for BMSCs [18,19]. As anticipated, the expression of CD29 (98.4%) and CD44 (99.7%) was positive, while that of CD11b (0.036%) and CD45 (0.084%) was negative in the mBMSCs used in this study (Figure 2).

### 2.3. The Expression of Circ-Spen Increases in Osteogenesis-Induced mBMSCs and Has Been Identified as a Circular RNA in the Nucleus

To probe the relationship between Circ-Spen and bone formation, mBMSCs were incubated in an osteogenic induction medium for 14 days and RNA was extracted on days 1, 3, 7, and 14. Over time, the expression level of Circ-Spen increased gradually (Figure 3A), which suggested that Circ-Spen played a crucial role in bone formation. Subsequently, to verify the characterization of Circ-Spen, divergent and convergent primers were designed to amplify the cDNA of Circ-Spen and linear *Spen*, after which the products were treated with or without RNase R [20]. Gel electrophoresis analysis demonstrated that the cDNA of *Spen* could be amplified by both divergent and convergent primers. However, the products amplified by convergent primers were completely digested by RNase R, while those amplified by divergent primers remained intact, which confirmed the circular structure of Circ-Spen (Figure 3B). It is located on chromosome 4:141,516,836–141,522,392 (804 nt), and the genomic structure suggests that Circ-Spen consists of the second and third exons from the *Spen* gene (Figure 3C). Additionally, the result of the nuclear/cytoplasm separation assay indicated that Circ-Spen was preferentially localized in the nucleus (Figure 3D). Thus, the stable expression cell strains of Circ-Spen NC, Circ-Spen, sh-NC, and sh-Circ-Spen were successfully established for subsequent investigations (Figure 3E, F). Circ-Spen was significantly higher-expressed in Circ-Spen than Circ-Spen NC, while it was downregulated in sh-Circ-Spen compared to sh-NC. Collectively, these results demonstrated that Circ-Spen was upregulated during the osteogenesis induction of mBMSCs and was identified as a circular RNA in the nucleus.

### 2.4. Circ-Spen Inhibits Cell Apoptosis in mBMSCs

Next, the effect of Circ-Spen on cellular apoptosis was investigated by MTT assays. The result showed that the cell viability of mBMSCs in the Circ-Spen group was similar to that in the Circ-Spen NC group, while it was notably decreased in the sh-Circ-Spen group compared with the sh-NC group, regardless of whether in 24 h or 48 h (Figure 4A,B). To further investigate whether the inhibitory activity of Circ-Spen was associated with apoptosis, the expression of signaling proteins related to apoptosis was tested by Western blotting in these cells. Accordingly, the level of antiapoptotic protein BCL-2 was remarkably elevated in the Circ-Spen group, while it was inhibited in the sh-Circ-Spen group. Additionally, the apoptotic proteins cleaved caspase-3/caspase-3 and BAX was significantly elevated in the sh-Circ-Spen group, while the expression of BAX was decreased in the Circ-Spen group (Figure 4C,D). In conclusion, these results demonstrated that Circ-Spen effectively inhibited cell apoptosis in mBMSCs.

### 2.5. Circ-Spen Promotes Autophagy in mBMSCs

The number of autophagosomes in mBMSCs was observed using a transmission electron microscope. As shown in Figure 5A, the number of autophagosomes in the Circ-Spen group was increased compared to the Circ-Spen NC group. However, the number of autophagosomes in the sh-Circ-Spen group was reduced compared to the sh-NC group. To further investigate the effect of Circ-Spen on autophagy in mBMSCs, whether autophagy markers changed with varying levels of Circ-Spen was examined. The result showed that BECN1 and P62 levels were elevated in the Circ-Spen group compared to the Circ-Spen NC group, while they were reduced in the sh-Circ-Spen group compared to the sh-NC group (Figure 5B, C). Taken together, these results indicated that Circ-Spen promoted autophagy in mBMSCs.

## 3. Discussion

Previously, circRNAs were considered to be by-products of molecular splicing with no discernible physiological function [21]. But in recent years, the development of high-throughput RNA-Seq and circRNA-specific computational tools has accelerated the identification and functional characterization of circRNAs [22]. Due to the tissue-specific and cell-specific expression patterns of circRNAs, research on circRNAs in BMSCs related to bone formation is also increasing. A study revealed that in rat BMSCs of glucocorticoid-induced osteoporosis (GIOP) models, 7 circRNAs were upregulated while 10 were downregulated compared to the control group, indicating the potential of circRNAs as biomarkers for bone health [23]. Furthermore, hsa_circ_0066523 has also been proven to increase the osteogenic induction process of hBMSCs, which promotes the proliferation and differentiation of hBMSCs by epigenetically repressing PTEN and thus activating the AKT pathway [24]. Similarly, in this study, Circ-Spen was identified as a circular RNA that increased during the osteogenic induction process of mBMSCs and was primarily expressed in the nucleus.

The cell viability of BMSCs plays a crucial role in both bone formation and skeletal tissue regeneration [25]. Thus, to investigate the mechanism of Circ-Spen in promoting bone health, cell viability and related protein expression levels of mBMSCs were measured in both the control and experimental groups. As we know, apoptosis is the most common form of programmed cell death, which can be triggered by a variety of physical, chemical, and biological factors, and its cellular responses are tightly regulated [26]. Furthermore, caspases are proteolytic enzymes and crucial effectors in apoptosis, among which the activation of caspase-3 (cleaved caspase-3) indicates cell apoptosis as a major member [27]. In this study, caspase-3 was not significantly activated in the Circ-Spen NC and Circ-Spen groups, while it was significantly activated in the sh-Circ-Spen group compared to the sh-NC group. Beyond that, a critical step in endogenous apoptosis involves the initiation of irreversible mitochondrial outer membrane permeability, which is regulated by members of the BCL-2 protein family [28]. The BCL-2 protein family comprises both pro-apoptotic and anti-apoptotic members. The anti-apoptotic members, such as BCL-2, inhibit apoptosis by impeding the release of mitochondrial apoptogenic factors, such as cytochrome c, into the cytoplasm to prevent apoptosis. Contrarily, pro-apoptotic members such as BAX facilitate apoptosis by releasing cytochrome c from the mitochondria, which triggers the apoptotic cascade [29]. It was reported that the apoptosis in IL-1β-stimulated-chondrocytes was reversed by knocking down circRNA_0092516 via inhibiting the expression of cleaved caspase-3 and BAX while promoting BCL-2 [30]. In addition, tumor necrosis factor-α was also reported to induce cell apoptosis of MC3T3 by increasing the expression of caspase-3, cleaved caspase-3, and BAX, which could be blocked by miR-14a silencing or co-culture with mBMSC-derived Exos [31]. Similarly, the downregulated expression of Circ-Spen in mBMSCs accelerated the apoptosis by promoting the expression of cleaved caspase-3 and BAX in this study while inhibiting BCL-2. Conversely, the overexpression of Circ-Spen could reverse this situation, which suggests the active role of Circ-Spen in regulating the cell viability of mBMSCs.

Both apoptosis and autophagy are related to cell death and regulate bone metabolism by determining the fate of bone cells [32]. In some cases, common upstream signals may trigger both autophagy and apoptosis, resulting in their co-occurrence [33]. It has been shown that in an oxidative stress microenvironment, the level of mitophagy decreases, leading to a massive accumulation of damaged mitochondria and cell apoptosis [7]. Another study also revealed that autophagy was essential in MSC-directed tissue regeneration by offering energy and metabolic precursors required for MSC differentiation and regulating the secretion of regenerative growth factors [34]. Therefore, the autophagy of mBMSCs was investigated in different groups and the number of autophagosomes in the Circ-Spen group was detected to increase but decrease in the sh-Circ-Spen group, indicating that autophagy in the Circ-Spen group was activated. BECN1 (Beclin 1) is a crucial protein in both the formation and maturation of autophagosomes, which acts as an adaptor to recruit multiple proteins modulating VPS34 [35]. Additionally, it has been suggested that the BH3 domain of BECN1 interacts with antiapoptotic proteins BCL-2 and BCL-XL, and this complex is involved in autophagy regulation [36]. In addition, serving as an autophagy substrate consisting of Atg5 and LC3, P62 regulates the formation of autophagosomes [37]. Similarly, P62 is considered a regulator in both autophagy and apoptosis [38]. The expression of BECN1 and P62 was also measured in this experiment, both of which significantly increased in the Circ-Spen group while decreasing in the sh-Circ-Spen group. All results indicated that Circ-Spen activated autophagy by increasing the expression levels of BECN1 and P62.

In conclusion, this study has revealed for the first time the active role of Circ-Spen in regulating the apoptosis and autophagy of mBMSCs by overexpressing and knocking down Circ-Spen. These findings reveal the potential of targeting Circ-Spen as a promising therapeutic strategy for rejuvenating senescent mBMSCs and enhancing the efficiency of mBMSC transplantation, which lays the foundation for advancements in the field of bone therapy.

## 4. Materials and Methods

### 4.1. Animals

Breeding pairs of *ERβ*^−/−^ mice (heterozygous male, heterozygous female, Cat# 026,176) maintained on the C57BL/6J background were generated from Jackson Labs. Three-month-old female C57BL/6J WT and *ERβ*^−/−^ mice were used in the experiment. All procedures for animal experiments were carried out by conforming to the requirements of the Animal Welfare Act, and all operations were approved by the Animal Care Committee of Jinan University.

### 4.2. RNA-Seq

After the mice were anesthetized, the two thighs were separated, the muscles and other tissues were carefully stripped away, and the femurs were retained. Any remaining tissues on the surface of the bone were gently wiped with gauze soaked in normal saline. Following this, to prevent degradation, RNA extraction was conducted after grinding the bone tissue with liquid nitrogen. Total RNA from the femurs of *ERβ*^−/−^ and WT mice was extracted using Trizol reagent (Invitrogen, Carlsbad, CA, USA) following the manufacturer’s protocol. Subsequently, the concentration and purity of the extracted RNA were assessed using Nanodrop (Thermo Fisher Scientific, Waltham, MA, USA). A complementary deoxyribonucleic acid library was constructed from the total RNA using a small RNA sample Pre Kit (Illumina, San Diego, CA, USA). Agilent 2100 Bioanalyzer (Agilent Technologies, Palo Alto, CA, USA) was used to check the integrity and size of cDNA. Lastly, the qualified cDNA was sequenced on a single-end HiSeq Xten platform (Illumina, San Diego, CA, USA). The differential expressions of circRNAs were shown in the heatmap (|Fold Change| ≥ 1 and *p* < 0.05) and identified by qRT-PCR. *Gapdh* was used as the internal reference. The primers (listed in Table 1) were designed and synthesized by Sangon Biotech (Shanghai, China).

### 4.3. Cell Culture

Cells were obtained from Procell (Wuhan, China) and were cultured in a 5% CO_2_ environment at 37 °C using DMEM (Thermo Fisher Scientific, Waltham, MA, USA) supplemented with 15% fetal bovine serum (CellMax, Peking, China). When the cell confluence reached 80–90%, mBMSCs were passaged or cryopreserved with trypsin (Solarbio, Peking, China) without EDTA. Only cells from 3 to 8 generations were utilized for this experiment.

### 4.4. Flow Cytometry Analysis

When mBMSCs at the third passage reached 80–90% confluency, they were digested and incubated with monoclonal antibodies specific to CD29 (Biolegend, San Diego, CA, USA), CD44 (Biolegend, San Diego, CA, USA), CD11b (Biolegend, San Diego, CA, USA), and CD45 (Biolegend, San Diego, CA, USA), and then analyzed by flow cytometry (FACSCanto, Becton, Dickinson and Company, Franklin Lake, NJ, USA).

### 4.5. Cell Transfection

Recombinant lentivirus including pHelper 1.0, pHelper 2.0 and Circ-Spen overexpression/knockdown plasmid (CMV-Circ-Spen-EF1-ZsGreen1-T2A-puromycin) (10^9^ TU/mL, MOI = 10) (Genechem, Shanghai, China), and HitransG P (Genechem, Shanghai, China) were incubated in a complete medium and added into mBMSCs at 20–30% confluency. After 16 h, the supernatant was replaced with a fresh complete medium. Unsuccessfully transfected cells were selected with 20 μg/mL puromycin (Beyotime, Shanghai, China) after 72 h, while successfully transfected cells were passaged and harvested for RNA extraction after 48 h and protein extraction after 72 h.

### 4.6. RNase R Treatment

Total RNA was extracted from mBMSCs by Trizol reagent. The total RNA was divided into two parts: one for the digestion of RNase R (RNase R+) and the other for the control group with a digestion buffer (RNase R−). For the first part, 2 μg of total RNA with 2 μL of 10 × RNase R Reaction Buffer and 2 μL of RNase R (20 U/μL, Epicentre Biotechnologies, Madison, WI, USA) were mixed; as for the second part, DEPC-treated water was used to replace RNase R. Subsequently, the RNA samples were incubated in 37 °C water for 30 min and then reverse-transcribed into cDNA. To verify the back-spliced events, convergent and divergent primers were designed and synthesized as follows (5′–3′): Circ-Spen (convergent) F: ATATGGCCGCGTGGAAAGTG, Circ-Spen (convergent) R: CTGTCAGTGTTGCTGCTGCTG, Circ-Spen (divergent) F: GCTCCAGGAGTCGATCCTCCA, Circ-Spen (divergent) R: GATGTCCACAAAATCCACAAAGGCA. RT-PCR was performed in ABI 7500 (Thermo Fisher Scientific, Waltham, MA, USA) and the products were verified using Southern blot.

### 4.7. CircRNA Localization

To localize Circ-Spen, a nucleoplasmic separation kit (BestBio, Nanjing, China) was used to isolate and extract the nuclear and cytoplasmic RNA of mBMSCs, and qRT-PCR was utilized to measure the expression of Circ-Spen in the above two parts. *Gapdh* served as cytoplasmic control, while *U6* (Sangon Biotech, Shanghai, China) acted as the nucleus control. The primers (listed in Table 1) were designed and synthesized by Sangon Biotech (Shanghai, China).

### 4.8. Transmission Electron Microscopy (TEM)

Cells were fixed in Glutaraldehyde Fix Solution (Servicebio, Wuhan, China) at 4 °C for 2–4 h. After being washed and centrifuged, the samples were put into 1% preheated agarose solution and incubated in 1% osmic acid at room temperature in a dark environment. They were embedded in SPI-Pon after dehydrating through a graded series of ethanol. Polymerized at 60 °C for 48 h, the samples were cut into 60–80 nm thick ultra-thin sections and stained with uranium acetate and lead citrate. Afterwards, images were acquired using HT7800 TEM (HITACHI, Tokyo, Japan).

### 4.9. Methylthiazol Tetrazolium (MTT) Assays

Cells were seeded into 96-well plates at a density of 3000 cells/well in triplicates and incubated overnight at 37 °C in a serum-free medium. The serum-free medium was then replaced by a complete medium, and the cells were cultured under the same conditions for 24 h and 48 h. Then, 20 μL MTT solution (5 mg/mL) (Beyotime, Shanghai, China) was added to each well and the supernatant was carefully discarded after 4 h of incubation. Then, the plates were shaken for 10 min after adding 150 μL DMSO (MP Biomedicals, Santa Ana, CA, USA). The absorbance of each well was measured using a microplate reader (Tecan Sunrise, Hombrechtikon, Switzerland) at 490 nm.

### 4.10. RNA Isolation and qRT-PCR

Cells were lysed in Trizol for 5 min at 4 °C. All of them were transferred to a 1.5 mL centrifuge tube and chloroform was added and mixed gently. After centrifugation at 4 °C for 15 min, isopropanol and ethanol were added to the supernatant in sequence. Total RNA was collected and its absorbance was measured using a NanoDrop 2000 spectrophotometer (Thermo Fisher Scientific, Waltham, CA, USA). Total RNA was reverse-transcribed into cDNA and qRT-PCR analysis was carried out in triplicate using an ABI7500. *Actb* was used as the internal reference. The primers (listed in Table 1) were designed and synthesized by Sangon Biotech (Shanghai, China). The cycling conditions consisted of 95 °C for 2 min followed by 40 cycles of 95 °C for 15 s and 60 °C for 30 s. Gene expression levels were quantified using the 2^−ΔΔCt^ method.

### 4.11. Western Blotting

Cells were lysed with RIPA lysis buffer (Beyotime, Shanghai, China) containing 10% PMSF (Beyotime, Shanghai, China) for 15 min and the lysate was centrifuged at 12,500 rpm for 15 min at 4 °C. After removing the supernatant, the protein concentration of each sample was measured using a BCA Protein Assay Kit (Invitrogen, Carlsbad, CA, USA). Equal amounts of protein per sample (10–20 μg) were loaded and separated on a 10% Glycine SDS-PAGE. Thereafter, the resolved proteins were subsequently electro-transferred onto 0.2 μm polyvinylidene difluoride (PVDF) membranes (Bio-Rad, Hercules, CA, USA) using a constant current of 300 mA. After being blocked with 5% (*w*/*v*) non-fat dried milk in Tris-buffered saline containing 0.1% Tween 20 (TBST) for 60 min at room temperature, the membranes were incubated overnight at 4 °C with the following antibodies: ACTB (4970S, Cell Signaling Technology, Danvers, MA, USA; 1:1000), cleaved caspase-3 (9664S, Cell Signaling Technology, Danvers, MA, USA; 1:1000), caspase-3 (ab184787, Abcam, Cambridge, UK; 1:1000), BCL-2 (ab59348, Abcam, Cambridge, UK; 1:1000), BAX (2772S, Cell Signaling Technology, Danvers, MA, USA; 1:1000), BECN1 (ab62557, Abcam, Cambridge, UK; 1:1000), P62 (16177S, Cell Signaling Technology, Danvers, MA, USA; 1:1000). The incubation was carried out in QuickBlock™ Primary Antibody Dilution Buffer for Western Blot (Beyotime, Shanghai, China). Subsequently, the membranes were then incubated with secondary antibodies that were appropriately conjugated to the horseradish peroxidase. The signals from the membranes were detected by the Bio-Rad Gel Doc XR System (Hercules, CA, USA) and the immunoreactive bands were quantified and analyzed using the Image J software (version 1.52a). The expression level of proteins was normalized to the ACTB control.

### 4.12. Statistical Analysis

SPSS 25.0 (SPSS Inc., Chicago, IL, NY, USA) was used to analyze the results above, which are expressed as means ± SD. The unpaired *t*-test was used for two-group comparison, and one-way ANOVA was applied to compare between three or multiple groups. A significance level of *p* < 0.05 was considered statistically significant.

## 5. Conclusions

In this study, the expression of Circ-Spen was markedly increased in *ERβ*^−/−^ mice femurs. Moreover, Circ-Spen exhibited an enhanced expression during the bone formation process of mBMSCs. Functionally, it inhibited apoptosis via caspase-3, BCL-2, and BAX, while also promoting autophagy through BECN1 and P62 in mBMSCs. These findings reveal the potential of targeting Circ-Spen as a promising therapeutic strategy for rejuvenating senescent mBMSCs and enhancing the efficiency of mBMSC transplantation, which lays the foundation for advancements in the field of bone therapy.

## Figures and Tables

**Figure 1 ijms-25-03034-f001:**
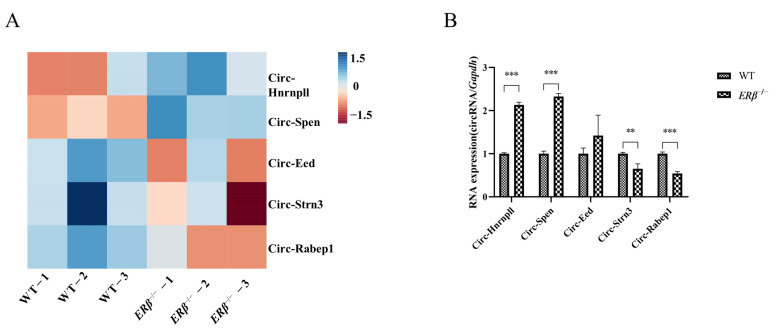
The differential expressions of circRNAs in *ERβ*^−/−^ mice. (**A**) Heatmap for RNA-Seq of the differential expression of circRNAs expressed in WT and *ERβ*^−/−^ mice femurs. (**B**) qRT-PCR was used to validate the different circRNAs of RNA-Seq in WT and *ERβ*^−/−^ mice femurs. Data are presented as mean ± SD (*n* = 3). ** *p* < 0.01; *** *p* < 0.001.

**Figure 2 ijms-25-03034-f002:**
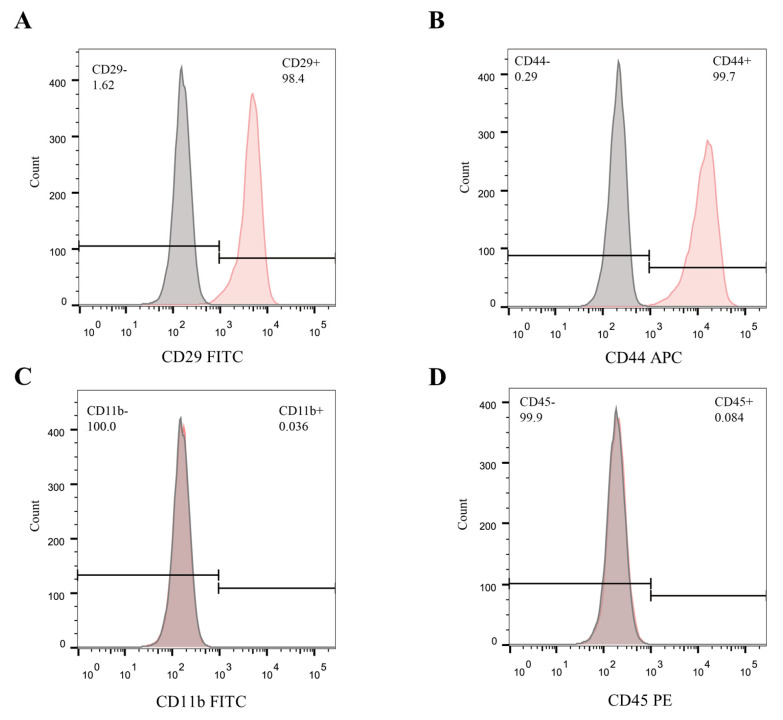
Identification of mBMSCs. (**A**–**D**) Flow cytometry analysis for mBMSCs incubated with fluorescently labeled antibodies against CD29, CD44, CD11b, and CD45.

**Figure 3 ijms-25-03034-f003:**
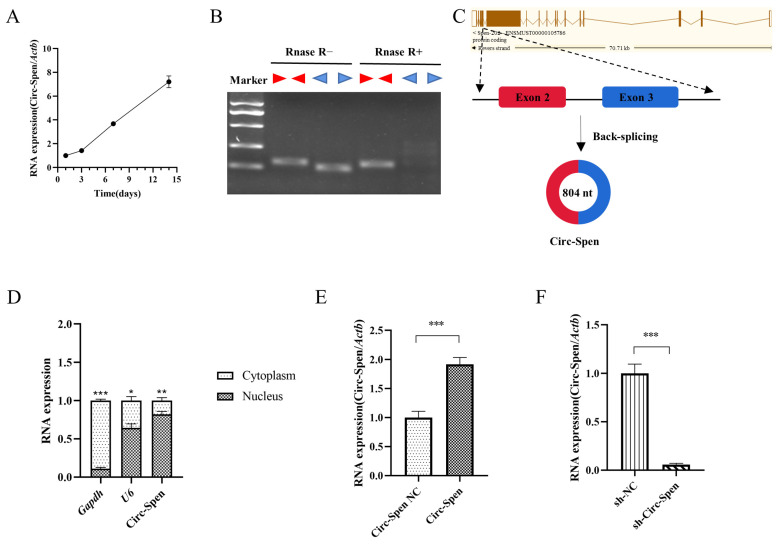
The expression of Circ-Spen increases during osteogenesis induction of mBMSCs and its characteristics are identified. (**A**) The expression of Circ-Spen was detected in mBMSCs cultured in osteogenic induction medium from day 1 to 14. (**B**) The circular nature of Circ-Spen was verified by RT-PCR and RNase R using divergent and convergent primers. (**C**) The genomic location and back-splicing pattern of Circ-Spen. (**D**) qRT-PCR was utilized to assess the levels of *Gapdh*, *U6*, and Circ-Spen in the cytoplasm or nucleus of mBMSCs. (**E**,**F**) qRT-PCR for the abundance of Circ-Spen in Circ-Spen NC, Circ-Spen, sh-NC, and sh-Circ-Spen cell strains. Data are presented as mean ± SD (*n* = 3). * *p* < 0.05; ** *p* < 0.01; *** *p* < 0.001.

**Figure 4 ijms-25-03034-f004:**
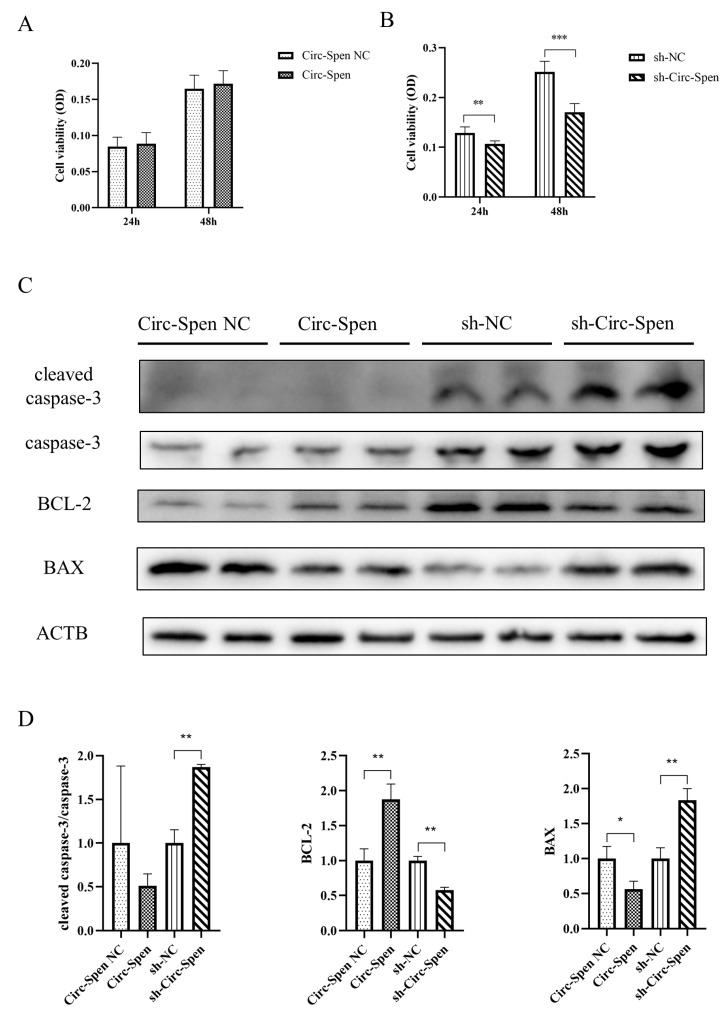
Circ-Spen inhibits cell apoptosis in mBMSCs. (**A**, **B**) The cell viability of mBMSCs in Circ-Spen NC, Circ-Spen, sh-NC, and sh-Circ-Spen groups was detected by MTT assays at 24 h and 48 h. (**C**) Western blotting analysis revealed the expression of cleaved caspase-3, caspase-3, BCL-2, and BAX. (**D**) The relative protein levels of cleaved caspase-3/caspase-3, BCL-2, and BAX were analyzed in indicated groups. The protein levels in the Circ-Spen NC group and the sh-NC group were used to standardize the relative protein ratio. The loading control was β-actin (ACTB). Data are presented as mean ± SD (*n* = 3). * *p* < 0.05; ** *p* < 0.01; *** *p* < 0.001.

**Figure 5 ijms-25-03034-f005:**
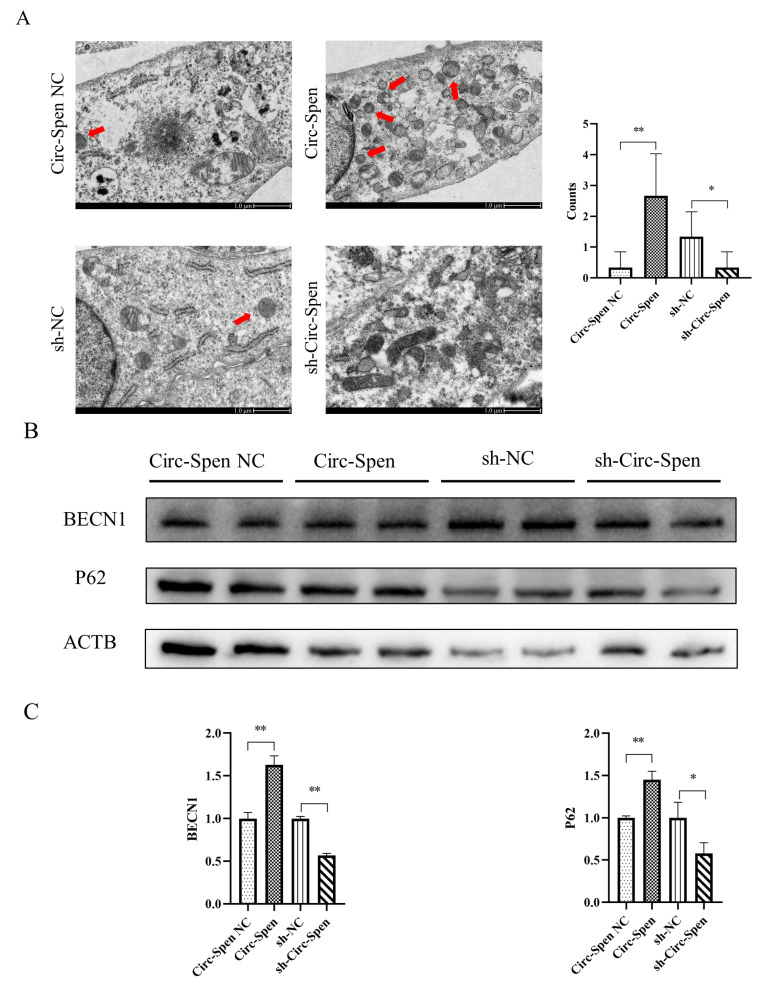
Circ-Spen promotes autophagy in mBMSCs. (**A**) Electron microscopy was performed to observe the autophagosome in each group, with the red arrow indicating its location. The scale bar measures 1.0 μm. The histogram plots represent autophagosome counts. Data are presented as mean ± SD (*n* = 6). (**B**) Western blotting analysis was conducted to assess the expression of BECN1 and P62. (**C**) The relative protein levels of BECN1 and P62 were analyzed in the indicated groups. The protein levels in the Circ-Spen NC group and the sh-NC group were used to standardize the relative protein ratio. The loading control was ACTB. Data are presented as mean ± SD (*n* = 3). * *p* < 0.05; ** *p* < 0.01.

**Table 1 ijms-25-03034-t001:** Primers for qRT-PCR.

Gene	Primer Sequence (5′–3′)
*Gapdh*	F: CAAGGCTGAGAACGGGAAG
	R: TGAAGACGCCAGTGGACTC
Circ-Spen	F: AGTCGATCCTCCAGTAGTGAC
	R: CAGCCACTCCTCCTTCAGAC
Circ-Hnrnpll	F: GAGATAGAGGAAAGGGTCGCC
	R: CACAACAGATTCACAGAGCCC
Circ-Eed	F: GATCCTCATAAAGCCAAGCCA
	R: GCCAGGTTTCCAGCATACAA
Circ-Strn3	F: CGGAGTTCAGGGGATGGTAC
	R: CTTGGTTCAGTTCTGTGCCATA
Circ-Rabep1	F: CGGGAAATAGCTGACTTAAGAAG
	R: CCAATTCTGCTACCCGTTGC
*Actb*	F: GCTGTGCTATGTTGCCCTAGACTTC
	R: GGAACCGCTCATTGCCGATAGTG

## Data Availability

All data used in the article can be found in Appendix A.

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
