# Peer review of "Bone-Differentiation-Associated Circ-Spen Regulates Death of Mouse Bone Marrow Mesenchymal Stem Cells by Inhibiting Apoptosis and Promoting Autophagy"

_ijms, 2024, doi:10.3390/ijms25053034_

Round 1
Reviewer 1 Report
Comments and Suggestions for Authors
It is an interesting article that presents a possible molecular target for therapies that promote bone regeneration considering the physiological variations associated with the action of a hormone. The introduction is adequate, the results are promising and consistent with the methodology used. The conclusion adequately summarizes the main results.
The following observations or suggestions arise from the analysis of the study.
Although the methodology has been adequately detailed, it would be a great contribution to specify the technique used to obtain tissue for molecular techniques considering a calcified or partially calcified structure such as the femur of a rat.
Although the results are consistent with the study methodology applied, it is necessary that the graphs in Figure 2 look better, more readable.
Sincerely,
The reviewer.
Author Response
Dear reviewer:
Thank you for your thorough examination and feedback on our article. We have addressed your comments and suggestions in the attachment. Kindly review our responses at your convenience. Thank you.

Reviewer 2 Report
Comments and Suggestions for Authors
In this article, authors demonstrated a differential expression of Circ-spen in ERb-/- mice vs wild type and they investigated the role of Circ-Spen in BM-MSCs osteogenesis and survival.
Please see below for my comments and suggestions.
Result section:
Figure 2
The quality of graphs are poor, it’s hard to read the labels. What do authors means by blank control? They should have used an un-stained and/or isotype control Abs as negative controls and I would suggest to over lay the histogram from the negative control and stained cells for each of the fluorophore to clearly show the difference between negative control and positively stained cells.
Figure 3:
Authors showed an increase in Circ-Spen expression in MSCs while cultured in an osteogenic differentiation medium. Consequently, they either over-expressed or knocked-down Circ-Spen in BM-MSC followed with assessment of apoptosis and autophagy. However, they didn’t include any data to investigate the effect of Circ-Spen modulation in osteogenic differentiation of BM-MSCs directly. I would suggest to culture the modified MSCs in osteogenic medium and assess the differentiation capacity in the modified cells vs naive MSCs.
Figure 5
A: It seems that there are more auto-phagosome in the image form sh-Cirs-Spen compared to sh-NC. How many fields have been imaged? Would it be possible to do any quantification on these images?
B: There are huge difference in intensity of the actin band? Do authors have the blots from another replicate of the experiment. They noted n=3 for this experiment in the figure legend.
General comment/question of the results
Although data from in vitro experiments (via over expression or dowregulation of Circ-Spen in BMS-MSC) pointed out to the regulation of apoptosis and autophagy by Cir-Spen. But, authors didn't perform any experiments on the bone samples from wild type and ERb-/- mice to assess the impact of elevated Circ-Spen level in ERb-/- mice (i.e. apoptotic pathway and autophagy). Additionally, do authors think that upregulation of Circ-Spen may affect the bone density.
Methods
Section 4.5
Authors should include details about the viral vectors used in their experiment (type, titer, efficiency of transduction, etc).
Author Response

(The authors gave the same response as above.)

Round 2
Reviewer 2 Report
Comments and Suggestions for Authors
I would like to thank authors for enhancing the quality of their manuscript by improving the results presentation and adding more details to the methodologies. They have also convincingly addressed my other comments regarding including more data. In my opinion, the new version of the manuscript is qualified to be published in IJMS.
Best regards.